# Facial Soft Tissue Thickness Differences among Three Skeletal Classes in Korean Population Using CBCT

**DOI:** 10.3390/ijerph20032658

**Published:** 2023-02-01

**Authors:** Eunseo Park, Jisuk Chang, Jongtae Park

**Affiliations:** 1Department of Oral Anatomy, College of Dental, Dankook University, Dankook Institute for Future Science and Emerging Convergence, Cheonan 31116, Republic of Korea; 2Department of Bio-Health Convergency Open Sharing System, Dankook University, Cheonan 31116, Republic of Korea; 3Department of Sport Management, College of Sports Science, Dankook University, Cheonan 31116, Republic of Korea; 4Mechanobiology Dental Medicine Research Center, Cheonan 31116, Republic of Korea

**Keywords:** facial soft tissue thickness, skeletal classes, cephalometry, malocclusion, forensic, forensic science, forensic art, facial reconstruction, CBCT, 3D

## Abstract

Studies related to facial soft tissue thickness (FSTT) have been conducted since the late 19th century. Soft tissue is any tissue in the body that is not hardened by ossification or calcification processes, such as bones and teeth; and varies according to sex, age, race, and nutritional status. Forensically, soft tissue thickness plays an important role in cases where a cadaver has no unique characteristics; and the remains cannot be identified through DNA analysis, fingerprints, or examination of dental records. Therefore, the results of the current study suggest that the average thickness of the three skeletal classes (i.e., straight, concave, and convex) should be used for face restoration and forensic art research. It is thought that the current study’s results will be invaluable in the fields of forensic science, forensic art, anthropology, and dentistry. As a result, gender differences were observed in all classes, and the facial tissue thickness in Korean adults differed according to gender and occlusion type.

## 1. Introduction

Studies related to facial soft tissue thickness (FSTT) have been conducted since the end of the 19th century. Soft tissue is any tissue in the body that is not hardened by ossification or calcification processes, such as bones and teeth. It varies according to gender, age, race, and nutritional status [1,2]. In addition, FSTT is forensic art tool used to identify individuals during face reconstruction [3]. Facial reconstruction and FSTT play a critical role in identifying both victims and perpetrators when corpses have no unique characteristics and their identities cannot be confirmed through more conventional methods, such as DNA analysis, fingerprint analysis, and the examination of dental records. In addition, FSTT can be useful in dental surgical diagnosis and treatment planning. Specifically, FSTT is often used for diagnosing and treating patients undergoing orthognathic surgery. However, to be able to accurately determine FSTT, it is necessary to know the average thickness of a specific part of the face [4,5,6,7]. Although many previous studies have investigated the relationship between FSTT and gender, age, race, and ethnicity, studies reporting FSTT in malocclusion cases are lacking.

Malocclusion refers to an occlusal relationship that is aesthetically and functionally problematic because the arrangement of teeth is not aligned for various reasons, or the state of the upper and lower meshing is out of the normal position [8]. Malocclusion cases are divided into dental and skeletal malocclusion. Dental malocclusion occurs when abnormalities manifest in the size, position, and shape of the teeth, while skeletal malocclusion occurs when the mandible and teeth are not in harmony due to skeletal causes, or when the relationship between the mandible is undesirable within the facial structure complex [9]. This study was conducted as a comparison study of FSTT measurements between different skeletal classes based on malocclusion patients’ facial profiles. Specifically, participating malocclusion patients’ facial profiles were divided into three skeletal classes according to their ANB angles: I-straight, II-convex, and III-concave. In orthodontic diagnosis and treatment, this classification of facial profiles is important in conducting facial recognition.

Various techniques for FSTT data collection have been proposed and tested by forensic scientists and anthropologists dating back from the 19th century up to the present [10]. According to Hwang et al. [11], by utilizing cone beam computerized tomography (CBCT), it is possible to make repeat measurements and confirm the landmark positions, and accurately measure FSTT. As a result, the utilization of CBCT imaging for FSTT measurements is spreading as well as analyses of CBCT data. Despite its popular use in the dentistry industry, studies using CBCT to measure FSTT have been lacking. Therefore, in this study, three-dimensional (3D) facial models were reconstructed based on CBCT data, and soft tissue was measured through 3D cephalometric measurements with a high accuracy. This allowed for the landmark identification and ease of FSTT measurement.

Cephalometric analysis is a method for measuring the size and spatial relationship between the size of the head and the size of teeth, mandible, and skull in orthodontics. It is frequently used as a treatment planning tool by dentists, orthodontists, and oral and maxillofacial surgeons [12]. It is also used to support forensic investigations. Moreover, previous studies use anatomical landmarks in cephalometric measurements. However, past studies have predominantly been conducted on Western subjects. Considering that race plays an important role in skeletal structure, studies on Asian subjects are much needed. Therefore, this study aims to present foundational data for the average thickness of FSTT according to skeletal class by using a 3D program that can compose CBCT data of Koreans that is highly similar to their actual faces.

## 2. Materials and Methods

The FSTT measurement subjects in this study were malocclusion 102 (male 48, female 54) patients in their 20s who visited the Dankook University Dental Hospital, had no missing teeth, asymmetry, or systemic diseases, and received CBCT data from the orthodontics department. Subjects were divided into the three skeletal classes, resulting in 16 subjects per class for the males and 18 subjects per class for the females. In addition, the number of study subjects was calculated through the G-power 3.1 (HHU, Dusseldorf, Germany) program. The CBCT data of this study were conducted after obtaining IRB (approval no. DKUDH IRB 2020-01-007) approval from the Dankook University Dental Hospital after including an application for consent exemption as a retrospective analysis.

All the scans were performed by the same technician, and the subjects were positioned so that the Frankfort Horizontal plane (FHP) was perpendicular to the floor to reduce the difference in FSTT. Photographs were taken after aligning the sagittal midline of the face with the imaging device. Afterward, the CBCT data was taken with a scanner (Alphard 3030, Asahi, Kyoto, Japan) and provided in DICOM format. Finally, the process values taken were carried out as an image scale of CT Scanning—slice increment, 0.39 mm; slice thickness, 0.39 mm; matrix, 512 px × 512 px.

The DICOM files taken by CBCT were 3D modeled for the patient’s hard and soft tissues in three directions using the Mimics (version 22.0, Materialise, Leuven, Belgium) 3D program. To extract hard and soft tissues in 3D, the HU (Hounsfield Unit) value, a numerical value representing the grayscale, was adjusted. In addition, the Hounsfield criterion was set according to the specified average range in the Mimics software to extract hard tissue and soft tissue.

To produce the hard-tissue 3D model, masking was performed by setting custom threshold values at a minimum (300~600) HU and a maximum (3071) HU. The hard-tissue mask was created by removing unnecessary parts using the edit mask function. For STL extraction, an object was created using the calculate part function.

To produce the soft-tissue 3D model, masking was performed by setting custom threshold values at a minimum (−500~300) HU and a maximum (225) HU. The soft-tissue mask was created by removing unnecessary parts using the edit mask function. For STL extraction, an object was created using the calculate part function.

The CBCT data were classified into 3 skeletal classes based on the ANB angle, which indicates the position of the maxilla in relation to the mandible. The 3 skeletal classes were classified as follows: Class I= 0° < ANB < 4°, Class II = ANB > 4°, Class III= ANB < 0°. The points assessed were: (A) the deepest point on the line between the anterior nasal spine (ANS) and the prosthion; (B) the deepest point from the line between the infradentale (the apex of the alveolar bone between the right and left lower first incisors) and the pogonion; and the nasion (Figure 1).

FHP was measured at 9 landmarks in the middle line of the anatomical landmark: 1. glabella (G); 2. nasion (N); 3. rhinion (Rhi); 4. subnasale (Sn); 5. labrale superius (Ls); 6. labrale inferius (Li); 7. labiomentale (Lbm); 8. pogonion (Pog); and 9. gnathion (Gn). Landmarks were perpendicular to the FHP or bone surface (Figure 2).

The glabella is the most prominent point in the median sagittal plane between the supraorbital ridges. The nasion is the most anterior point on frontonasal suture. The rhinion is the tip of the nasal bone. The subnasale is in the middle, the junction where the base of the columella of the nose meets the upper lip. The labrale superius is the point denoting the vermilion border of the upper lip. The labrale inferius is the point denoting the vermilion border of the lower lip in the midsagittal plane. The labiometale is the most concave point on the mandibular symphysis. The pogonion is the most anterior point of the mandibular symphysis. The gnathion is the point located perpendicular on the mandibular symphysis, midway between the pogonion and menton.

The most central sagittal reconstruction was selected, and the measurements of soft tissue thickness were performed in 9 different landmarks in the midline region, following the methodology of Utsuno et al. [13]. Measurements were evaluated by calculating the average value of repeated measurements (i.e., each measurement was performed more than twice) performed by researcher Park and Professor Park, respectively. Measures of central tendency (mean and standard deviation, maximum and minimum values) were calculated for each soft tissue measurement.

Measurement items were analyzed using the SPSS (version 23.0, IBM Corporation, Armonk, NY, USA) statistical package. After the normality of the data was confirmed, the difference between the three skeletal classes was compared via the Kruskal-Wallis test. Differences between genders were compared between each skeletal class through the Mann–Whitney test. The post-hoc test was conducted with a 95% confidence interval, and the significance level was set at 0.05.

## 3. Results

Table 1 depicts the mean thickness (mm), range (minimum, maximum value), and standard deviation (SD) values of the Korean male population classified according to the three skeletal classes. The results indicate that males showed a significant difference in Li. A difference was found in one out of nine landmarks. Among the three classifications, class II and class III were thicker at four landmarks each among the three skeletal classes. The average values were thicker at landmarks G in class I; Li, Lbm, Pog, and Gn in class II; and N, Rhi, Sn, and Ls in class III.

Table 2 depicts the mean thickness (mm), range (minimum, maximum value), and standard deviation (SD) values of the Korean female population classified according to the three skeletal classes. As a result of the measurement, women showed significant difference in G, N, Sn, and Li. Differences were found in four out of nine landmarks. Among the three skeletal classes, five and four landmarks were thicker in class I and class III, respectively. The average values were thicker at N, Rhi, Lbm, Pog, and Gn in class I; Li in class II; and G, N, Sn, and Ls in class III.

Table 3 depicts the mean thickness (mm) and standard deviation (SD) values by gender of the Korean population classified according to the three skeletal classes. The measurement results showed significant results for N and Sn in class I; N, Rhi, Li, Lbm, and Gn in class II; and N, Rhi, Sn, Ls, and Li in class III. Among the classifications, it can be seen that there is a difference in thickness, especially in class II and class III. In particular, there were significant results between genders in N and Sn common to all three skeletal classes. FSTT was especially thick at the Li compared to other landmarks in class I, where male (15.36, SD: 1.92) and female (14.94, SD: 1.65) values were obtained. Similarly, in class II, the Li landmark was thicker than other landmarks (male: 19.13, SD: 3.23; female: 16.4, SD: 2.32). In class III, the Ls landmarks FSTT was thicker than other landmarks (male: 16.26, SD: 2.83; female: 14.21, SD: 1.45). There were differences in the lip area landmarks between the skeletal classes and genders, and between races.

## 4. Discussion

The present study indicates the presence of differences in thickness among skeletal classes between the genders, and males’ FSTT to be thicker than females at most landmarks. Especially for female subjects, these differences were observed in the upper lip region (labrale superius). Soft tissue can develop proportionally or disproportionately to the skeletal structure, depending on the muscle, fat, and skin covering the face [14]. Therefore, to accurately reconstruct a face, it is necessary to know the average FSTT of a specific part of the face. Through FSTT, forensic artists use artistic techniques to visualize crime, investigation, or information, one of which is face reconstruction [15]. Face reconstruction is a technique for examining postmortem faces from human remains, providing ancestry, age, gender, and facial profile [16,17,18]. In addition, FSTT is related to dental malocclusion, and research is underway. Among them, skeletal classes are classified using the ANB angle to quantify the relationship between the maxilla and mandible. However, most FSTT studies have been conducted on Westerners; thus, preliminary data on diverse ethnicities are insufficient.

In this study, we intend to build a data base for face types. Therefore, to accurately measure FSTT, CBCT was utilized due to its minimal distortion and ability to show three-dimensional cross-sections and volumes. In addition, CBCT used FHP and FSTT differences between skeletal classes were observed using CBCT data, and were processed via Mimics software. As a result, there were statistically significant features for each skeletal class category.

As a result of the FSTT comparison of Korean males especially in this study, males were thicker than females at all the landmarks in class II; and thicker than females at all the landmarks, except for G and Pog in III. According to Stephan CN et al. [19], the FSTT observed that men were thicker than women and also reported that men were thicker regardless of race. The reason why men are thicker is that men have larger bones and muscles than women [19]. According to previous studies, women’s skin generally lacks collagen synthesis, and hyaluronic acid synthesis is promoted by estrogen; however, men’s skin can become thicker because testosterone promotes collagen synthesis [2]. It is thought that it is necessary to measure soft-tissue thickness according to gender to improve the aesthetic part during orthodontic treatment.

Results of FSTT comparisons amongst female participants across skeletal classes returned significant differences in the G, N, Sn, and Li landmarks. Moreover, there were no significant differences in Rhi, Ls, Lbm, Pog, and Gn. These results differ from previous studies. For example, Utsuno et al. [13] found significant differences between classes at landmarks Sn, Ls, Sto, Lbm, and Pog. However, their study was based on Japanese participants and did not have malocclusion. According to Formby et al. [20], the soft tissue in women can change after their mid-twenties due to hormonal changes that follow pregnancy and childbirth. Furthermore, it has been reported that the individual’s nutritional status and environmental factors can be influential because facial morphology represents a complex interaction between genetics and the individual’s environmental surroundings [21]. Based on the past literature, such extraneous variables may have been the cause for the inconsistency with Utsuno et al.’s [13] results. Therefore, futures studies on FSTT should incorporate additional variables, such as age, childbirth status, and other environmental factors.

Finally, the comparisons between genders indicated that males had thicker soft tissue around the lips than females. This was similar to the results of other previous studies conducted on Koreans. For instance, Hwang et al.’s [22] reported that the part corresponding to the lip area was observed to be thicker in men than in women. In addition, when comparing the current study’s Ls measurements of Korean women with Utsuno et al.’s [13] results of Japanese women, our study results indicated that Korean women’s Ls were thicker. Korean women’s Ls results were class I (14.19), class II (13.71), and class III (16.52); and Japanese women’s results were class I (12.53), class II (11.88), and class III (14.03). However, the current study did not directly measure Japanese women; and thus, conclusions should be made with caution. The reason for such a large difference around the lips is that it does not change where soft tissue is attached to the bone; however, it has been reported that the oral cavity and teeth have a lot of movement; thus, it dramatically affects the soft-tissue depth [23,24]. This observation is thought to affect the thickness due to the protrusion of the mandible for each class. The previous study differed in the number of subjects from this study. Utsuno et al.’s [13] study subjects consisted of with 38 in class I, 22 in class II, and 43 in class III, which may be the reason for the different study results. In addition, according to the Oliver et al.’s [25] study, it was observed that the correlation between mandibular bossing and soft tissue change was more pronounced in patients with thin lips compared to patients with thick lips. Based on these findings, our study also found significant gender differences in each skeletal class. Given that the FSTT around the mouth seems to differ between races, this phenomenon warrants future forensic and clinical studies.

## 5. Conclusions

The current study found that there were clear differences between the genders in FSTT across skeletal classes. Differences were most pronounced in the upper- and lower-lip areas, where males had thicker soft tissue compared to females in all the skeletal classes. In conclusion, there were differences in the FSTT of Korean adults according to gender and occlusion type. These results give support to the notion that the average thickness of skeletal classes should be used in the fields of forensic art, forensic medicine, anthropology, medical, and dentistry. In addition, given that the average thickness of the FSTT is related to various ages and growth states, and the characteristics of each gender vary by the skeleton, further studies on FSTT are needed.

## Figures and Tables

**Figure 1 ijerph-20-02658-f001:**
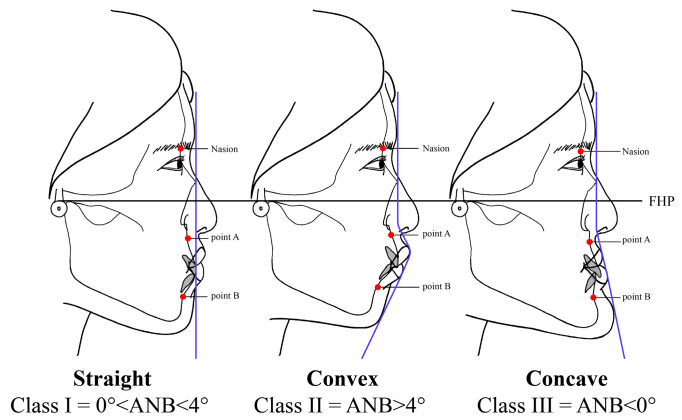
Division and profile of the 3 skeletal classes.

**Figure 2 ijerph-20-02658-f002:**
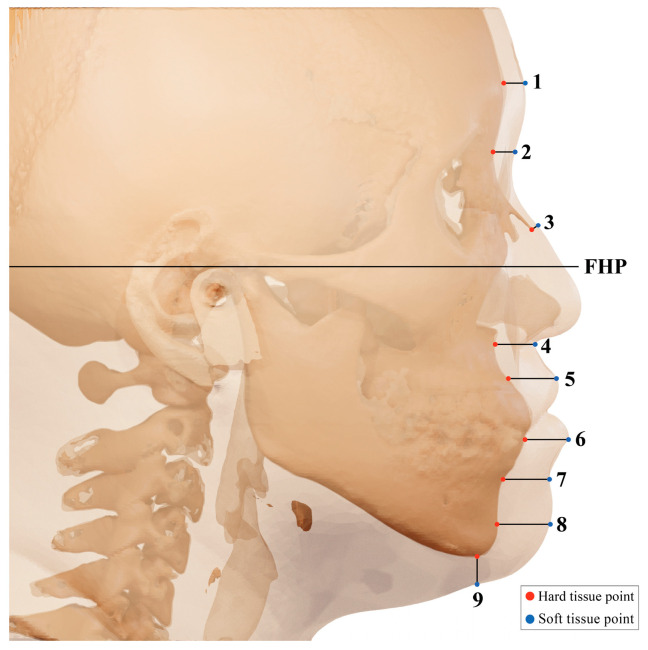
Location of measurement points for facial soft tissue thickness: 1. glabella; 2. nasion; 3. rhinion; 4. subnasale; 5. labrale superius; 6. labrale inferius; 7. labiomentale; 8. pogonion; 9. gnathion.

**Table 1 ijerph-20-02658-t001:** The differences between three skeletal classes in FSTT in Korean males.

Landmark	Class I (N = 16)			Class II (N = 16)			Class III (N = 16)			
	Mean (SD)	Min	Max	Mean (SD)	Min	Max	Mean (SD)	Min	Max	*p*-Value
G	5.76 (0.76)	4.68	7.43	5.32 (0.72)	4.30	7.02	5.52 (0.65)	4.30	6.65	NS
N	6.93 (0.97)	5.07	8.24	6.54 (0.86)	5.46	8.97	7 (0.89)	5.07	8.61	NS
Rhi	2.92 (0.82)	1.66	4.44	2.85 (1.2)	0.56	6.23	3.32 (0.69)	2.49	5.01	NS
Sn	15.25 (1.52)	13.26	17.94	14.58 (2.15)	10.58	17.93	16.26 (1.79)	12.51	19.50	NS
Ls	15.16 (2.4)	12.09	19.49	14.42 (2.72)	11.31	20.30	16.52 (2.83)	11.69	19.89	NS
Li	15.36 (1.92)	12.50	18.34	19.13 (3.23)	14.81	25.73	14.43 (1.69)	11.69	17.16	**
Lbm	11.92 (1.28)	9.36	13.66	13.34 (1.82)	10.92	16.01	12.29 (1.56)	9.74	16.77	NS
Pog	11.27 (2.6)	7.80	14.83	11.99 (2.35)	7.80	15.20	11.73 (1.44)	8.98	13.64	NS
Gn	7.22 (1.6)	5.07	9.76	8.42 (2.3)	5.47	12.85	7.56 (2.02)	5.06	12.50	NS

Data are M-sample mean (sample standard deviation); *p*-values were obtained by the Kruskal-Wallis test, ** *p* < 0.001, NS = No Significance.

**Table 2 ijerph-20-02658-t002:** The differences between three skeletal classes in FSTT in Korean females.

Landmark	Class I (N = 18)			Class II (N = 18)			Class III (N = 18)			
	Mean (SD)	Min	Max	Mean (SD)	Min	Max	Mean (SD)	Min	Max	*p*-Value
G	5.57 (0.68)	4.30	7.03	5.03 (0.64)	3.88	6.23	5.84 (0.67)	4.67	7.03	*
N	6.16 (1.48)	4.28	9.77	5.25 (0.72)	4.29	6.64	6.16 (1.12)	3.90	7.81	*
Rhi	3.10 (1.6)	1.41	6.63	2.41 (0.45)	1.65	3.05	2.77 (1.41)	1.41	7.59	NS
Sn	13.53 (1.2)	10.59	15.17	12.29 (1.24)	10.14	14.83	13.93 (1.62)	11.73	16.83	*
Ls	14.19 (1.69)	11.28	16.37	13.71 (2.18)	10.52	17.54	14.21 (1.45)	11.70	16.39	NS
Li	14.94 (1.65)	11.73	17.94	16.40 (2.32)	11.31	20.28	13.20 (1.92)	10.54	17.94	**
Lbm	11.99 (0.92)	10.54	14.82	11.48 (1.82)	8.58	14.83	11.15 (1.33)	8.19	13.27	NS
Pog	12.44 (1.83)	7.81	14.83	11.51 (2.23)	8.19	15.22	11.98 (2.5)	7.80	16.37	NS
Gn	7.53 (1.92)	4.70	11.32	6.21 (1.24)	4.29	8.19	7.50 (2.23)	4.69	12.11	NS

Data are M-sample mean (sample standard deviation); *p*-values were obtained by the Kruskal-Wallis test, * *p* < 0.05 ** *p* < 0.001, NS = No Significance.

**Table 3 ijerph-20-02658-t003:** Mean, SD of Korean population (mm) among three skeletal classes.

Landmark		Class I		Class II		Class III	
	Gender (N)	Mean (SD)	*p*-Value	Mean (SD)	*p*-Value	Mean (SD)	*p*-Value
G	M (16)	5.76 (0.76)	NS	5.35 (0.72)	NS	5.52 (0.65)	NS
	F (18)	5.57 (0.68)		5.03 (0.64)		5.84 (0.67)	
N	M (16)	6.93 (0.97)	*	6.54 (0.86)	**	7.00 (0.89)	*
	F (18)	6.16 (1.48)		5.25 (0.72)		6.16 (1.12)	
Rhi	M (16)	2.92 (0.82)	NS	2.85 (0.56)	NS	3.32 (069)	*
	F (18)	3.10 (1.60)		2.41 (0.45)		2.77 (1.41)	
Sn	M (16)	15.25 (1.52)	*	14.58 (2.15)	**	16.26 (1.79)	**
	F (18)	13.53 (1.20)		12.29 (1.24)		13.93 (1.62)	
Ls	M (16)	15.16 (2.4)	NS	14.42 (2.72)	NS	16.52 (2.83)	*
	F (18)	14.19 (1.69)		13.71 (2.18)		14.21 (1.45)	
Li	M (16)	15.36 (1.92)	NS	19.13 (3.23)	*	14.43 (1.69)	*
	F (18)	14.94 (1.65)		16.4 (2.32)		13.2 (1.92)	
Lbm	M (16)	11.92 (1.28)	NS	13.34 (1.82)	*	12.29 (1.56)	NS
	F (18)	11.99 (0.92)		11.48 (1.82)		11.15 (1.33)	
Pog	M (16)	11.27 (2.6)	NS	11.99 (2.35)	NS	11.73 (1.44)	NS
	F (18)	12.44 (1.83)		11.51 (2.23)		11.98 (2.5)	
Gn	M (16)	7.22 (1.60)	NS	8.42 (2.30)	*	7.56 (2.02)	NS
	F (18)	7.53 (1.92)		6.21 (1.24)		7.50 (2.23)	

Data are M-sample mean (sample standard deviation). *p*-value were obtained by Kruskal Wallis test, * *p* < 0.05 ** *p* < 0.001, NS = No Significance.

## Data Availability

Original data are available upon request to the corresponding author.

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
