# Peer review of "Facial Soft Tissue Thickness Differences among Three Skeletal Classes in Korean Population Using CBCT"

_ijerph, 2023, doi:10.3390/ijerph20032658_

Round 1

Reviewer 1 Report

It is very interesting subject to study, and I personally congratulates the authors. However, the author compare to western studies published but do not discuss a major difference between caucasian vs. orientals which is the distribution of hair follicles over the face and lips in western male. It will be interested to read about it in the revision 

Author Response

Hello this is researcher Park.

Thank you for your commets.

I searched for several articles, but they were not relevant to my article, so I focused on malocclusion and gender rather than differences on race. But I'm also curious about the research on what you said. If you think it should be added to this paper, could you recommend a related article?

Reviewer 2 Report

Thank you very much for letting me review this paper. The manuscript is well written and certainly deserves publication. The authors have collected rare data that is of great use in today's forensic and clinical practise. I recommend publication after the following minor and major revisions:

1. In the introduction, when referring to 4.5.6, use commas instead.

2. In the "Material and Method" section, please define the following:

- age range, as age is one of the factors affecting FSTT

- missing teeth - does this mean extracted, unerupted or not developed at all?

3. I think that each measurement should be repeated by more than one observer or at least by one operator twice, in order to improve the statistics and make the results more accurate and reliable.

3. What study was used to determine the landmarks and what study was used to divide them into three skeletal cases?

4. Explain/define the ANB angle.

5. review the definition of the ANB angle in the straight and convex classes in Figure 1 - for both there are same values.

6. Why did you use/run nonparametric tests such as Kruskal Wallis and Mann Whitney if you state that the normality distribution of the data was confirmed (lines 132-133, Material and Method section)?

7. mean, SD, minimum, and maximum should be referred to as "measures of central tendency and dispersion".

8. The titles of Tables 1 and 2 should be changed according to the results they provide, e.g., 1 and 2 show the differences between skeletal classes in FSTT in Korean men/women, then Table 3 shows the intersexual differences in FSTT in the Korean population according to the three skeletal classes.

9. What are the results of post-hoc tests if significant differences were present in Tables 1 and 2?

10. In the discussion section, the first paragraph should be a summary of the main results and their interpretations. Then, the authors should discuss age and position (supine, upright) as factors affecting FSTT, e.g., study limitations.

11. Also, such a general statement should not be made in the conclusion part, line 244. Based on your results, only differences between biological sexes and between three skeletal stages were studied. To make such a statement as yours, the best regressors or correlates should be tested/examined first.

12. I suggest including the CBCT method in the title, as well as malocclusion, since there are studies examining, for example, B-mode ultrasound, participants without malocclusion, and others.

Author Response

Hello, this is researcher Park.

Thanks for commenting on my research.

Edited based on reviewer comments, and data were uploaded as word files.

Reviewer 3 Report

"Measurements were evaluated by calculating the average value of repeated measurements by the researcher and Professor Park respectively." This line was unclear. Please clarify who the observers were.

Why did the authors do the measurements via Mimics from the STL object, instead of directly measuring from the CBCT DICOM data via MPR view?

Many CBCT machines stabilizes patient's head with a chin rest/chin cup. If the chin is pressed against the chin rest hardly, the soft tissue around the chin will be distorted. Please elaborate in the Methods how the authors avoided this problem for their prospective scans.

Author Response

(The authors gave the same response as above.)

Round 2

Reviewer 3 Report

The authors have satisfactorily addressed my concerns.